# Comparative Grain Chemical Composition, Ruminal Degradation In Vivo, and Intestinal Digestibility In Vitro of *Vicia Sativa* L. Varieties Grown on the Tibetan Plateau

**DOI:** 10.3390/ani9050212

**Published:** 2019-05-02

**Authors:** Yafeng Huang, Rui Li, Jeffrey A. Coulter, Zhixin Zhang, Zhibiao Nan

**Affiliations:** 1State Key Laboratory of Grassland Agro-ecosystems, Lanzhou University, Lanzhou 730020, China; huangyafeng316@163.com (Y.H.); lir15@lzu.edu.cn (R.L.); zhixinzhang.phd@gmail.com (Z.Z.); 2Key Laboratory of Grassland Livestock Industry Innovation of Ministry of Agriculture and Rural Affairs, Lanzhou University, Lanzhou 730020, China; 3College of Pastoral Agriculture Science and Technology, Lanzhou University, Lanzhou 730020, China; 4Department of Agronomy and Plant Genetics, University of Minnesota, St. Paul, MN 55108, USA; jeffcoulter@umn.edu

**Keywords:** common vetch, grain, nutritive value, ruminants

## Abstract

**Simple Summary:**

Common vetch (*Vicia sativa* L.) grain is an important source of protein in rations for ruminants, but little information is available on the protein value of common vetch grains, both in terms of chemical composition and protein degradability, and regarding variation between intra-species and year. The objective of this study was to evaluate grain chemical composition, ruminal protein degradability in vivo, and intestinal protein digestibility in vitro of four common vetch varieties over two cropping years on the Tibetan Plateau. This study was also conducted to establish correlations of grain chemical composition with ruminal degradability parameters of grain protein and with intestinal digestibility of grain protein. Results of this study demonstrated that grain quality characteristics varied significantly among varieties and years. The relationship between grain chemical composition and intestinally absorbable digestible protein (IADP) was best described by a linear regression equation, and coefficients of determination remained very high (*R*^2^ = 0.891). Overall, the results indicated that in terms of effective crude protein degradability and IADP of grain, common vetch varieties Lanjian No. 2 and Lanjian No. 3 have the greatest potential among varieties examined for supplementing ruminant diets when grown on the Tibetan Plateau.

**Abstract:**

Four varieties of common vetch, three improved varieties and one local variety, were evaluated for grain chemical composition, rumen protein degradability, and intestinal protein digestibility over two cropping years on the Tibetan Plateau. This study also examined correlations of grain chemical composition with rumen degradability parameters of grain protein and with intestinal digestibility of grain protein. Results of this study showed that grain quality attributes varied (*p* < 0.05) among varieties and cropping years. Significant intra-species variation was observed for concentrations (g/kg dry matter) of crude protein (CP; range = 347–374), ether extract (range = 15.8–19.6), neutral detergent fiber (aNDF; range = 201–237), acid detergent fiber (range = 58.2–71.6), ash (range = 27.6–31.0), effective CP degradability (EDCP; range = 732–801 g/kg CP), and intestinally absorbable digestible protein (IADP; range = 136–208 g/kg CP). The relationship between grain chemical composition and IADP was best described by the linear regression equation IADP = –0.828CP + 8.80ash + 0.635aNDF + 70.2 (*R*^2^ = 0.891), indicating that chemical analysis offers a quick and reliable method for IADP of common vetch grain. In terms of EDCP and IADP of grain, common vetch varieties, Lanjian No.2 and Lanjian No. 3, have the greatest potential among varieties tested for supplementing ruminant diets when grown on the Tibetan Plateau.

## 1. Introduction

Livestock sector is the largest land-use system worldwide and is an engine of economic growth that has received significant attention within the last decade [1,2]. The Tibetan Plateau covers an area of 256 million ha^2^ and is the largest grassland ecosystem in Eurasia [3]. It supports nearly 41 million Tibetan sheep (*Ovies aries*) and 14.3 million yak (*Bos grunniens*), cattle (*Bos taurus*), and cattle-yak hybrids, which are a predominant part of the local economy for the nomadic population of over 9.8 million [3]. The Tibetan Plateau is also the source of Asia’s major river systems, including the Mekong, Yangtze, Yellow, Ganges, and Indus. Therefore, sustainable development and management of the Tibetan Plateau is of vital importance, not only for supporting the livelihood of millions of people, but also for protection of these critical river systems. 

Legume grains are important sources of protein in ruminant nutrition [4,5,6]. Demand for plant-based protein has grown steadily, since the influence of the bovine spongiform encephalopathy which resulted in the banning of meat meal for feeding ruminants in the European Union in 1994 [7]. The Tibetan Plateau has an inherently extreme and unstable climate, and a short growing season, which seriously limits crop production [8], especially for leguminous crops [9]. Nan et al. [9] reported that many annual legume crops [e.g., woolly-pod vetch (*Vicia villosa* ssp. *dasycarpa Roth*) have low grain yield potential in this region. Common vetch was introduced to the Tibetan Plateau of China in 1998 and has shown promising potential for grain production [9]. Recently, new varieties of common vetch designated as Lanjian No.1, Lanjian No.2, and Lanjian No.3, which represent diverse levels of maturity with acceptable grain production on the Tibetan Plateau, were developed by the common vetch (*Vicia sativa* L.) breeding program at Lanzhou University [10].

Common vetch is a cool-season annual legume cultivated in many countries that can satisfy grain demand for feed and food with low requirement for nitrogen input due to biological nitrogen fixation [5,11]. The nutritive value of common vetch grain is relatively high, with an average of 314 g/kg crude protein (CP) and 961 g/kg CP total digestible protein [7,12,13]. This makes common vetch grain a potentially important source of protein in rations for ruminants, such as cattle and sheep, and is of great interest in mixed crop-livestock systems [5,7,13,14,15], particularly due to the banning of soybean (*Glycine max* (L.) Merr.) meal for feeding organic livestock [16]. Koumas and Economides [14] and Gül et al. [15] reported satisfactory ruminant growth with diets containing common vetch grain as a replacement for soybean (*Glycine max* (L.) Merr.) meal. 

For the feed application of common vetch grain, previous studies focused on grain chemical composition [4,11,17] or nutritional values of a single variety [7]. The nutrient value of common vetch for ruminants varies widely in the literature and depends on the common vetch variety and animal species, and chemical composition of common vetch grain. These phenomena indicate the need to analyze the grain of common vetch varieties for nutritive value before using in ration formulations. However, little information is available on protein value of common vetch grains, regarding variation between intra-species and the growing season. Rumen degradability of protein is a key parameter for the evaluation of feed protein value in ruminants [7,18], and the benefits of protein resources are dependent on protein digestion in the small intestine [7,19]. Minerals are a minor ingredient of ruminant feeds, but they are essential for growth and development [20]. Existing data on the mineral composition of common vetch grain is limited. Woods et al. [21,22] reported that chemical analysis of concentrate feedstuffs could be used for accurate prediction of ruminal and intestinal protein degradability. These observations could be more cost and time effective than using in situ nylon bags and in vitro techniques with surgically prepared animals, which are relatively expensive, labor-intensive, and subject to animal welfare problems [21,22]. However, no available information has been reported on the relationship between grain chemical composition with grain rumen degradation parameters of CP and grain intestinal protein digestibility for common vetch. 

Understanding the varietal differences in nutritive value of common vetch grain grown on the Tibetan Plateau could increase its use as a protein supplement in ruminant diets and lead to greater integration of common vetch within agro-pastoral farming systems in this region, thereby reducing the need for protein feed imports and enhancing the income of farmers. Against this background, the objectives of this study were: (i) to evaluate the potential of common vetch for use as a protein feed source in ruminant diets, by studying grain chemical composition, ruminal degradability kinetics of CP, and intestinal protein digestibility of four common vetch varieties over two cropping years; (ii) to investigate relationships between grain protein, fiber fractions, and ash concentrations, and grain ruminal degradability parameters of CP and intestinally digestible protein of common vetch varieties grown in rainfed conditions on the Tibetan Plateau.

## 2. Materials and Methods 

### 2.1. Experimental Site

Common vetch was planted in rainfed conditions during the 2015 and 2016 cropping seasons at the Xiahe experimental station of Lanzhou University, China (35°19′ N, 102°58′E; 2880 m above sea level (asl)), which is located on the eastern margin of the Tibetan Plateau. The soil type of the site is classified as chernozem and is slightly acidic (pH 6.8), low in phosphorous (0.73 g/kg) but adequate in potassium (13.5 g/kg). The preceding crop was rape (*Brassica campestris* L.) in both years. The daily meteorological data (air temperature and rainfall) were recorded via an automatic meteorological station (PC200W, Campbell Scientific) installed near the experimental field. During the cropping period (April through September) in 2015 and 2016, mean air temperature was 10.8 and 12.0 °C, respectively, and total precipitation was 314 and 417 mm, respectively. Monthly mean temperature and total precipitation in July and August were 2.1 to 3.4 °C, respectively, and 22 to 34 mm greater in 2016 than 2015 (Table 1). 

### 2.2. Plant Material, Experimental Design, and Sampling

Three improved varieties of common vetch (Lanjian No. 1, Lanjian No. 2, and Lanjian No. 3) and one local variety were evaluated in a field experiment. The local variety originated in the Gansu province of China and is now widely cultivated in Gansu, Qinghai, and provinces located in the middle and lower reaches of the Yangtze River in China. These varieties were selected because they are grown extensively in the region surrounding the research site [10]. Agronomic characteristics of these varieties are in Table 2 [23]. Plots were planted at 150 viable seeds m^−2^, with four replicates per variety in a completely randomized design. Each plot was 8 × 5 m and contained 26 rows of plants, spaced 20 cm apart. Common vetch was planted on 6 May 2015 and 28 April 2016 following rhizobial inoculation (CCBAU01069, China Agricultural University, Beijing, China), which was recommended based on the symbiont performance for these varieties [10]. Irrigation and fertilizer were not applied to the experimental plots during either growing season and weeds were controlled by hand. 

At the pod maturity stage of common vetch, two representative 1 × 1 m sections from each plot were manually harvested and threshed to obtain grain samples. Grain was oven-dried at 60 ℃ for 48 h and then ground to pass through a 2-mm sieve to obtain samples for analyses of ruminal incubation and intestinal digestibility, and to pass through a 1-mm sieve to obtain samples for chemical analysis.

### 2.3. Chemical Analysis

Laboratory dry matter (DM) concentration was measured from 2.0 g of ground grain from each sample by drying in a forced-air oven at 135 ℃ for 2 h (method 930.15; AOAC) [24]. Subsequently, grain ash concentration was measured by incineration in a muffle furnace at 550 ℃ for 5 h (method 938.08; AOAC). Total N was measured using the Kjeldahl method (method 988.05) [24] and CP concentration was calculated as total N × 6.25. Ether extract (EE) concentration was measured by extraction with petroleum ether (method 920.85; AOAC) [24]. Concentrations of acid detergent fiber (ADF) and neutral detergent fiber (aNDF) were measured by sequential analysis with α-amylase and sodium sulfite, and expressed with residual ash excluded [25]. Mineral concentrations were determined by atomic absorption spectroscopy (method 985.01; AOAC) [26] and phosphorus (P) concentration was determined by colorimetry (method 965.17; AOAC) [24]. All measurements were performed in triplicate and chemical standards were included in each analytical run as appropriate. 

### 2.4. Ruminal Degradability of CP

Four adult fistulated dorper sheep rams 30–31 months of age and 58 ± 1 kg live body weight were used to determine ruminal degradation of CP of common vetch grain using the nylon bag technique described by Nandra et al. [27]. Briefly, 5.0 g of dry weight grain from each sample (in duplicate) was placed in a nylon bag (9 × 5 cm; 50-μm pore size) and incubated for 0 (control), 2, 4, 8, 12, 24, and 48 h in the rumen of four adult fistulated dorper rams. After removal, the bags were washed thoroughly with tap water and frozen (−20 °C) until further analysis. Prior to analysis, all bags were defrosted and manually washed with tap water until the water ran clear, before being oven-dried at 65 ℃ for 48 h and weighed. The dried undigested residues of replicates per time within rams were pooled prior to analysis. Degradability parameters of CP were determined using the exponential model described by Ørskov and McDonald [28]. The effective degradable fraction of CP was calculated as effective CP degradability (EDCP) = *A* + [(*B* × *C*)/(*C* + *k*)], where *A* is the soluble fraction of grain CP, *B* is the potentially degradable fraction of grain CP, *C* is the rate of degradation of fraction *B* (h^−1^), and *k* is the rumen outflow rate (0.031 h^−1^) [7]. The animals were housed in individual stalls and daily fed 1200 g of 550 g/kg DM sheepgrass (*Leymus chinensis*, (Trin.) Tzvel), 294 g/kg DM maize (*Zea mays* L.) grain, 140 g/kg DM soybean meal, and 16 g/kg DM of a concentrate mix at maintenance energy level [7] in two equal meals at 08:30 am and 16:30 pm. The rams had free access to water and mineral/vitamin licks. The experimental protocols were approved by the Animal Ethics Committee of Lanzhou University (protocol number: AEC-LZU-2016-01).

### 2.5. Intestinal Digestibility of CP

A modified 3-step in vitro procedure described by Gargallo et al. [29] was used to determine intestinal digestibility of rumen undegradable protein (IDP). Briefly, dried duplicate undegradable residues from the 16 h in situ ruminal incubation were pooled and ground to pass through a 1-mm sieve. Six sub-samples of dried sample residues (500 mg each) were then weighed into Ankom F57 filter bags and heat-sealed. Twenty-four sample bags were incubated in each incubation jar of a Daisy^II^ incubator (ANKOM Technology, Fairport, NY, USA) containing 2 L of 0.1 M pre-warmed HCl solution (pH = 1.9) and 1 g L^−1^ of pepsin (P-7000, Sigma, St. Louis, MO, USA), with constant rotation at 39 °C for 1 h. After incubation and washing, sample bags were reintroduced into the same incubation jars containing 2 L of pre-warmed pancreatin (0.5 M KH_2_PO_4_ buffer, pH = 7.75, 50 mg/kg of thymol, and 3 g/L of pancreatin (P-7545, Sigma)) and incubated with constant rotation at 39 °C for 24 h. After removal, bags were rinsed with tap water until the water ran clear, dried at 55 °C for 48 h, and reweighed. Undegradable residues from ruminal and intestinal incubations were analyzed for nutrient concentration. 

### 2.6. Statistical Analyses

Ruminal undegradable protein (RUP) concentration was calculated as 1000 – EDCP (g/kg CP), and intestinally absorbable digestible protein (IADP) was determined as RUP (g/kg of CP) × IDP (g/kg of RUP), as described by Lawrence and Anderson [30]. Total digestible protein (TDP) was calculated as TDP = EDCP + IADP. Statistical analyses were performed using SPSS software (Version 21.0. IBM Corporation, Armonk, NY, USA). Data were analyzed using analysis of variance to assess the significance of the main effects of common vetch variety and year, and their interaction. When the F-tests were significant, variances among means were compared using the Duncan significant difference test at *p* < 0.05. Pearson’s correlation analysis was used to evaluate relationships between chemical composition variables and *A*, *B*, *C*, and IADP of grain samples across varieties and years (*n* = 32). Chemical composition data of grain samples of common vetch varieties across years (*n* = 32) were analyzed to predict the rumen degradability parameters of CP (*A*, *B*, and *C*) and IADP using stepwise multiple linear regression [21,22].

## 3. Results

### 3.1. Chemical Composition

The effects of year and variety significantly influenced all proximate composition variables of common vetch grain with the exception of grain ash concentration, which was only affected by variety (Table 3). The interaction between year and variety did not significantly affect proximate composition variables. Averaged across varieties, CP concentration was less and EE, aNDF, and ADF concentrations were greater in 2016 compared to 2015. Averaged over both years, CP concentration among varieties ranged from 353 g/kg DM for Lanjian No. 3 to 370 g/kg DM for the local variety. Crude protein concentration of Lanjian No.1 was less than that of the local variety, but greater than that of the other improved varieties. Inversely, EE ranged from 16.1 g/kg DM for the local variety to 19.0 g/kg DM for Lanjian No. 3. The local variety had significantly less EE concentration compared to Lanjian No. 2 and Lanjian No. 3 but was similar to Lanjian No. 1. Averaged across years, aNDF and ADF among varieties ranged from 207 to 229 g/kg DM and 59.5 to 68.8 g/kg DM, respectively. Neutral detergent fiber and ADF of the local variety were not significantly different than that of Lanjian No. 1 but were less than that of other improved varieties. Ash concentration ranged from 27.8 g/kg DM for the local variety to 31.0 g/kg DM for Lanjian No. 3. The local variety had significantly less ash concentration compared to the improved varieties. There were significant differences in ash concentration among improved varieties, which was greater for Lanjian No. 2 and Lanjian No. 3 than Lanjian No. 1. 

Phosphorus (P), calcium (Ca), magnesium (Mg), and iron (Fe) concentrations in common vetch grain varied significantly between years and among varieties, but were not affected by the year × variety interaction (Table 3). Averaged across varieties, grain concentration of all measured macronutrients was greater in 2016 than in 2015. Means over both years, P, Mg, Ca, and Fe concentrations among varieties ranged from 1.25 to 3.42, 3.18 to 3.85, 1.39 to 1.95, and 0.431 to 0.801 g/kg DM, respectively. Phosphorus and Fe concentrations were lower and Mg and Ca concentrations were greater in grain of the local variety compared with the improved varieties, and there were no significant differences among improved varieties. Micronutrient levels also varied among varieties, but were not significantly affected by year or the interaction between year and variety. Averaged across years, grain concentrations of zinc (Zn), manganese (Mn), and copper (Cu) among varieties ranged from 36.3 to 48.6, 12.2 to 14.3, and 6.41 to 9.59 mg/kg DM, respectively. Micronutrient concentrations of grain of the local variety were greater than those of the improved varieties, and there were no significant differences among improved varieties. 

### 3.2. Ruminal Degradation Kinetics of CP

Soluble fraction of grain CP of common vetch was significantly influenced by year and variety, but not by their interaction (Table 4). Averaged across varieties, *A* was greater in 2016 than 2015 (346 and 325 g/kg, respectively). Averaged across years, *A* among varieties ranged from 312 g/kg for Lanjian No. 2 to 359 g/kg for the local variety, and *A* of the local variety did not differ significantly from that of Lanjian No. 1 and was greater than that of other improved varieties. There were significant differences in the *B* and *C* among varieties, although the effect of year and of year × variety interaction were not significant. Averaged across years, *B* among varieties ranged from 552 g/kg for Lanjian No. 3 to 608 g/kg for the local variety. Potentially degradable protein of the local variety was greater than that of Lanjian No. 2 and Lanjian No. 3, and did not differ significantly from that of Lanjian No.1. In contrast, *C* ranged from an average of 0.0770 h^−1^ for the local variety and Lanjian No. 1 to an average of 0.0966 h^−1^ for Lanjian No. 2 and Lanjian No. 3. The rate of protein degradation of the local variety was not significantly different from that of Lanjian No. 1, but was less than that of other improved varieties.

Effective CP degradability of common vetch grain differed significantly between years (772 and 755 g/kg CP in 2015 and 2016, respectively) and among varieties, but was not significantly influenced by the year × variety interaction. Averaged across years, EDCP among varieties ranged from 736 g/kg CP for Lanjian No. 3 to 792 g/kg CP for the local variety. The EDCP of the local variety was significantly greater than that of Lanjian No. 2 and Lanjian No. 3, but did not differ significantly from that of Lanjian No. 1.

### 3.3. Intestinal Digestibility of CP

The IDP and IADP of grain were significantly affected by year and variety of common vetch, but not by the year × variety interaction (Table 4). Averaged across varieties, IDP and IADP were less in 2015 than in 2016. Averaged across years, IDP and IADP among varieties ranged from 692 to 768 g/kg of RUP, and 144 to 203 g/kg CP, respectively. The IDP and IADP of the local variety were less than that of Lanjian No. 2 and Lanjian No. 3, but not significantly different than that of Lanjian No. 1. Total digestible protein of grain was not significantly affected by year, variety of common vetch, or the interaction between year and variety, and averaged 937 g/kg CP across years and varieties.

### 3.4. Prediction of A, B, C, and IADP

Correlations between grain chemical composition and *A*, *B*, *C*, and IADP were all significant at *p* < 0.05, with absolute values of the correlation coefficient ranging from 0.350 to 0.787 (Table 5). Therefore, stepwise multiple linear regression was used to predict *A*, *B*, *C*, and IADP. Goodness of fit of the regression models was evaluated based on the coefficient of determination (*R*^2^) and root mean squared error (RMSE). For each parameter, the selected regression model produced the highest *R*^2^ and lowest RMSE, with *R*^2^ values of 0.805, 0.712, 0.748, and 0.891 for prediction of *A*, *B*, *C*, and IADP, respectively (Table 6).

## 4. Discussion

### 4.1. Chemical Composition 

In this study, CP concentration of common vetch grain was greater in 2015 than 2016, while concentrations of EE, aNDF, and ADF were less in 2015 compared to 2016. Dornbos and Mullen [31] reported that drought stress during grain filling of soybean increased grain protein concentration and decreased lipid concentration, and Al-Karaki and Ereifej [32] reported lower protein and greater EE concentration of field pea (*Pisum sativum* L.) with greater precipitation during pod filling. Ítavo et al. [33] reported that EE concentration of legume grain is positively associated with energy release. These observations help to explain differences in the grain chemical composition between years in this study, which may be attributed to greater mean air temperature and total precipitation during July and August of 2016 when common vetch was in flowering to pod-filling stages of phenological development.

In this study, all varieties produced relatively high CP, with values ranging from 353 to 370 g/kg DM. These CP values were greater than those reported from Syria (266–316 g/kg DM) [11] and Turkey (249–279 g/kg DM) [34], but less than that of varieties grown in Jordan (mean = 393 g/kg DM) [12]. Neutral detergent fiber (207–229 g/kg DM) and ADF (59.5–68.8 g/kg DM) concentrations of grain of all varieties fell within the range of values for aNDF and ADF reported by Ramos-Morales et al. [7], Seifdavati and Taghizadeh [13], and Huang et al. [5] (aNDF = 144–423 g/kg DM, ADF = 52–124 g/kg DM). Ether extract of grain of common vetch varieties in this study (16.1–19.0 g/kg DM) was greater than that reported from Spain (14.3 g/kg DM) [7], but within the range reported by Karadag and Yavus [34] in Turkey (11.6–32.3 g/kg DM). Such differences in the proximate composition of grain of common vetch among studies may be attributed to variation in varieties, site, growing environment, and crop management [4,11,34]. The relative high levels of grain protein for all common vetch varieties evaluated in this study demonstrate that they could be a valuable supplement to low-quality ruminant diets, particularly for the local variety [7]. 

Macronutrient concentrations of common vetch grain in this study were greater in 2016 than 2015, which may be partially due to greater precipitation and warmer air temperature during grain filling in 2016, resulting in accelerated translocation of macronutrients from vegetative tissues and roots to the developing grain. Similarly, Thavarajah et al. [35] reported that warmer air temperature during grain filling was associated with greater Fe concentration of lentil (*Lens culinaris* L.) grain. Uzun et al. [36] reported that 1000-seed weight of common vetch grain was negatively correlated with most mineral elements, such as Ca, Mg, and Mn. Previous studies of these varieties tested showed that seed weight of the local variety was less than that the improved varieties (Table 2) [23]. In this study, large varietal differences in grain concentrations for most minerals may be attributed to a starch dilution effect in lager seeds [36]. Macro and micronutrient concentrations of grain of common vetch varieties in this study are within the ranges reported from studies in Turkey [36] and China [17]. Concentrations of P, Mg, Ca, and Fe in grain of common vetch grown in Cyprus [37] were 5.7, 1.70, 1.40, and 0.14 g/kg DM, respectively, in contrast to those in this study (mean = 2.77, 3.37, 1.59, and 0.69 g/kg DM, respectively). Grain concentrations of Cu and Mn in the present study are similar to values reported for common vetch grown in Jordan [12], although the concentration of Zn was greater in the present study. Grain Zn concentration in this study was similar to that of common vetch grown in Cyprus [37], but the levels of Cu and Mn were comparatively greater in that study. Differences in grain mineral concentrations among studies may be related to differences in varieties planted [36], the stage of grain maturity at harvest [12], and other factors such as site conditions, growing environment, and agronomic practices [17]. 

Overall, the large varietal differences in chemical composition of common vetch grain indicate that farmers can integrate grain of these varieties into ruminant diets in terms of quality traits to meet the nutrient demands of diverse livestock classes. 

### 4.2. Ruminal Degradability of CP

In this study, the values for *A* and *B* are in accordance with earlier reports on grain of common vetch [7,38], pea, and lupin (*Lupinus albus* var. *multolupa*) [7,39]. The average value for *C* (0.0867 h^−1^) of grain of the common vetch varieties in this study were in line with the previous reports on common vetch (0.081 h^−1^) as well as soybean meal (0.082 h^−1^) [38]. Mean EDCP of common vetch grain in this study (764 g/kg CP) was less than that (857 g/kg CP) reported from Greece studies by Zagorakis et al. [40], but in agreement with earlier observations from Spain (753 g/kg CP; [41]. Such differences in the EDCP of common vetch grain could be ascribed in part to methodological differences such as animal species [42], basal diet [38], rumen outflow rate [39], and protein availability among varieties. 

Effective CP degradability of grain was relatively high for all varieties in this study (736–792 g/kg CP). Therefore, these varieties could be considered as a source of degradable protein for ruminal microorganisms and then for microbial protein synthesis [7]. Zagorakis et al. [40] reported an optimum rumen degradable: undegradable protein ratio of 60:40 in feeds for highly productive ruminants. As a consequence, results of the present study suggest that if the tested varieties are used as a protein source in highly productive ruminant diets, they should first be treated to reduce CP degradability, such as with heat treatment [39]. Seifdavati and Taghizadeh [13] reported that autoclaved common vetch grain had 7.07% less EDCP compared to the raw grain. 

### 4.3. Intestinal Digestibility of CP

In this study, large variability in grain IDP among years and varieties (683–782 g/kg RUP) may be partially related to varietal differences in concentrations of ash and ADF. Fu et al. [43] reported that IDP of concentrated feedstuffs is positively associated with ADF concentration, while it was negatively associated with ash concentration. Mean IDP (731 g/kg RUP) of common vetch varieties in this study was greater than that reported for other varieties (600 g/kg RUP by Ramos-Morales et al. [7] and 668 g/kg RUP by Seifdavati and Taghizadeh [13]), but less than that reported for soybean meal (905 g/kg RUP) [30]. These differences among studies of common vetch may be ascribed to differences in common vetch varieties. 

In ruminants, the nutritive value of a protein supplementation depends on protein digestion in the small intestine [7,19]. In the present study, IADP of the local variety was less than that of Lanjian No. 2 and Lanjian No. 3, but did not differ significantly from that of Lanjian No. 1. This implies that Lanjian No. 2 and Lanjian No. 3 may be more suitable sources of protein in highly productive ruminant diets than the local variety and Lanjian No. 1. Reports on IADP in grain of common vetch are scarce. However, grain IADP of common vetch varieties in this study was similar to that reported for other annual grains such as camelina (*Camelina sativa* (L.) Crantz; 191 g/kg CP) [30], but lower than for soybean meal (376 g/kg CP) [30]. 

### 4.4. Prediction of Degradability Parameters and Intestinal Digestibility of Grain

A significant positive or negative correlation was observed between chemical composition and *A*, *B*, *C*, and IADP of common vetch grain in this study. Limited information has been reported on these relationships for the grain of legumes. To our knowledge, the information available is limited, with the exception of that from studies in Ireland, where concentrations of ash, CP, EE, NDF, ADF, and acid detergent lignin were significantly correlated with degradability parameters of CP in 12 concentrated feedstuffs [21,22].

The use of the chemical composition of concentrated feedstuffs to predict the degradability parameters and intestinal digestibility was previously proposed by Woods et al. [21,22]. Woods et al. [21,22] indicated that *R*^2^ values greater than 0.90 are desirable for equations predicting degradability parameters and intestinal digestibility of concentrated feeds. In the present study, the best fit equations for predicting *A*, *B*, and *C* based on chemical composition had *R*^2^ values of 0.805, 0.712, and 0.748, respectively. Therefore, the equations developed from this study may not be sufficiently precise to accurately predict degradability parameters based on chemical analyses. For equations predicting IADP, the *R*^2^ value increased from 0.787 when only CP was included in the prediction equation to 0.891 when CP, ash, and aNDF were included. Since the *R*^2^ value of this equation is close to 0.9, it may be feasible to use these variables, derived from chemical analysis, to predict grain IADP of common vetch. This result may be incorporated into future breeding programs for improving grain quality of common vetch.

## 5. Conclusions

All common vetch varieties evaluated in this study produced high grain CP concentration. However, when grain of these varieties is used as a protein source in diets of highly productive ruminants, it should be pre-treated to reduce ruminal degradability. This study also shows that CP, ash, and aNDF can be used to predict IADP of common vetch grain. Additionally, large varietal differences in nutritive value in this and other studies demonstrate that it is necessary to analyze grain of common vetch varieties for nutritive value before using in ration formulations. In terms of grain EDCP and IADP, the varieties Lanjian No. 2 and Lanjian No. 3 have great potential for supplementing ruminant diets when grown on the Tibetan Plateau.

## Figures and Tables

**Table 1 animals-09-00212-t001:** Monthly mean air temperature and total precipitation during the 2015 and 2016 cropping seasons at the Xiahe experimental station, Gansu, China.

	Mean Air Temperature (°C)		Total Precipitation (mm)	
Month	2015	2016	2015	2016
April	4.4	6.2	23	25
May	9.3	9.3	38	72
June	12.4	12.7	49	54
July	13.4	15.5	69	103
August	14.2	17.6	62	84
September	11.2	10.6	73	79
Sum			314	417

**Table 2 animals-09-00212-t002:** Agronomic characteristics of the common vetch varieties used in this experiment.

Agronomic Characteristic	Local Variety	Lanjian No.1	Lanjian No.2	Lanjian No.3
Days to mature (d)	134	145	132	124
1000 grains weight (g)	54	79	71	76
Plant height (cm)	92	106	80	69
Altitude (m.a.s.l)	–	<3000	<3500	<4000
Year of release	1987	2014	2015	2011

Source: Ministry of Agriculture, Beijing, China [23].

**Table 3 animals-09-00212-t003:** Effects of cropping year, variety, and the year × variety interaction on the chemical composition of four common vetch grain varieties grown on the Tibetan Plateau.

Dependent Variable	Mean across Years					Year										*p*-Value		
						2015					2016							
	Local Variety	Lanjian No.1	Lanjian No.2	Lanjian No.3	SEM ^1^	Local Variety	Lanjian No.1	Lanjian No.2	Lanjian No.3	SEM	Local Variety	Lanjian No.1	Lanjian No.2	Lanjian No.3	SEM	Year	Variety	Year × Variety
Proximate composition (g/kg DM)																		
CP	370 ^a^	360 ^b^	354 ^c^	353 ^c^	1.57	374 ^a^	365 ^b^	358 ^b^	358 ^b^	1.88	366 ^a^	356 ^b^	350 ^c^	347 ^c^	1.96	< 0.001	<0.001	0.827
EE	16.1 ^c^	17.5 ^bc^	18.5 ^ab^	19.0 ^a^	0.278	15.8 ^b^	17.2 ^ab^	17.8 ^a^	18.5 ^a^	0.372	17.3 ^b^	17.8 ^ab^	19.3 ^a^	19.6 ^a^	0.368	0.011	0.002	0.818
aNDF	207 ^c^	217 ^bc^	225 ^ab^	229 ^a^	2.46	201 ^b^	209 ^ab^	220 ^a^	220 ^a^	2.87	214 ^b^	225 ^ab^	231 ^a^	237 ^a^	3.19	< 0.001	<0.001	0.942
ADF	59.5 ^b^	61.3^b^	68.3^a^	68.8 ^a^	1.12	58.2 ^b^	59.0 ^ab^	65.8 ^a^	66.1 ^a^	1.38	60.8 ^c^	63.5 ^bc^	70.8 ^ab^	71.6 ^a^	1.63	0.014	<0.001	0.929
Ash	27.8 ^c^	29.3 ^b^	30.7 ^a^	31.0 ^a^	0.261	27.9 ^c^	29.6 ^b^	30.7 ^ab^	31.0 ^a^	0.353	27.6 ^c^	29.1 ^b^	30.8 ^a^	31.0 ^a^	0.395	0.548	<0.001	0.845
Macronutrient minerals (g/kg DM)																		
P	1.25 ^b^	3.06 ^a^	3.42 ^a^	3.33 ^a^	0.192	0.993 ^b^	2.78 ^a^	3.24 ^a^	3.08 ^a^	0.268	1.50 ^b^	3.33 ^a^	3.61 ^a^	3.57 ^a^	0.270	0.042	<0.001	0.993
Mg	3.85 ^a^	3.20 ^b^	3.18 ^b^	3.23 ^b^	0.0736	3.75 ^a^	3.05 ^b^	3.09 ^b^	3.11 ^b^	0.108	3.96 ^a^	3.35 ^b^	3.28 ^b^	3.35 ^b^	0.0947	0.047	<0.001	0.989
Ca	1.95 ^a^	1.39 ^b^	1.54 ^b^	1.47 ^b^	0.0655	1.90 ^a^	1.20 ^b^	1.36 ^b^	1.29 ^b^	0.105	2.00 ^a^	1.58 ^b^	1.72 ^ab^	1.65 ^b^	0.0612	0.007	0.003	0.396
Fe	0.431 ^b^	0.793 ^a^	0.801 ^a^	0.750 ^a^	0.0479	0.345	0.630	0.635	0.600	0.0467	0.517 ^b^	0.955 ^a^	0.967 ^a^	0.900 ^a^	0.0684	< 0.001	0.002	0.820
Micronutrient minerals (mg/kg DM)																		
Zn	48.6 ^a^	38.4 ^b^	37.7 ^b^	36.3 ^b^	1.30	50.0 ^a^	39.2 ^b^	38.7 ^b^	40.2 ^b^	1.76	47.2 ^a^	37.7 ^b^	36.7 ^b^	32.5 ^b^	1.88	0.097	<0.001	0.625
Mn	14.3 ^a^	12.2 ^b^	13.2 ^b^	13.1 ^b^	0.220	14.0	12.2	13.3	14.1	0.291	14.6 ^a^	12.2 ^b^	13.0 ^b^	12.0 ^b^	0.332	0.201	0.002	0.056
Cu	9.59 ^a^	6.81 ^b^	6.68 ^b^	6.41 ^b^	0.357	10.1 ^a^	7.79 ^ab^	6.30 ^b^	6.05 ^b^	0.589	9.10 ^a^	5.83 ^b^	7.05 ^b^	6.77 ^b^	0.420	0.527	0.002	0.271

^a,b^ Within a row, different letters represent the significant differences at *p*-value < 0.05 by the Duncan significant difference test. ^1^ SEM, standard error of the mean; aNDF, neutral detergent fiber; ADF, acid detergent fiber; Ca, calcium; CP, crude protein; Cu, copper; DM, dry matter; EE, ether extract; Fe, iron; Mg, magnesium; Mn, manganese; P, phosphorus; Zn, zinc.

**Table 4 animals-09-00212-t004:** Effects of cropping year, variety, and the year × variety interaction on the ruminal degradation kinetics of crude protein and intestinal protein digestibility of four common vetch varieties grown on the Tibetan Plateau.

Dependent Variable	Mean across Years					Year										*p*-Value		
						2015					2016							
	Local Variety	Lanjian No.1	Lanjian No.2	Lanjian No.3	SEM ^1^	Local variety	Lanjian No.1	Lanjian No.2	Lanjian No.3	SEM	Local Variety	Lanjian No.1	Lanjian No.2	Lanjian No.3	SEM	Year	Variety	Year × Variety
*A*, g/kg CP	359 ^a^	352 ^a^	312 ^b^	320 ^b^	5.03	367 ^a^	360 ^a^	327 ^b^	331 ^b^	6.13	352 ^a^	343 ^a^	298 ^b^	309 ^b^	7.22	0.004	<0.001	0.873
*B*, g/kg CP	608^a^	604 ^a^	570 ^b^	552 ^b^	5.86	610 ^a^	610 ^a^	565^ab^	537 ^b^	10.3	606 ^a^	597 ^ab^	575 ^bc^	567 ^c^	5.88	0.508	<0.001	0.355
*C*, h^−1^	0.0766 ^b^	0.0774 ^b^	0.0975 ^a^	0.0956 ^a^	0.00202	0.0727 ^b^	0.0759 ^b^	0.0981 ^a^	0.0985 ^a^	0.00321	0.0761 ^b^	0.0788 ^b^	0.0968 ^a^	0.0928 ^a^	0.00256	0.560	<0.001	0.587
EDCP, g/kg CP	792 ^a^	782 ^a^	744 ^b^	736 ^b^	5.02	801 ^a^	793 ^a^	755 ^b^	739 ^b^	7.38	782 ^a^	771 ^a^	732 ^b^	733 ^b^	6.30	<0.001	<0.001	0.514
IDP, g/kg of RUP	692 ^b^	706 ^b^	756 ^a^	768 ^a^	6.73	683 ^b^	705 ^b^	746 ^a^	754 ^a^	8.39	702 ^b^	708 ^b^	765 ^a^	782 ^a^	10.3	0.023	<0.001	0.514
IADP, g/kg CP	144 ^b^	154 ^b^	194 ^a^	203 ^a^	4.99	136 ^b^	146 ^b^	183 ^a^	197 ^a^	7.00	153 ^b^	162 ^b^	205 ^a^	208 ^a^	6.70	<0.001	<0.001	0.661
TDP, g/kg CP	936	937	937	939	1.03	937	940	938	936	1.12	935	934	937	942	1.76	0.700	0.861	0.796

^a,b^ Within a row, different letters represent the significant differences at *p*-value < 0.05 by the Duncan significant difference test. ^1^ SEM, standard error of the mean; *A*, soluble fraction of grain CP; *B*, potentially degradable fraction of grain CP; *C*, rate of degradation of fraction *B*; CP, crude protein; EDCP, effective CP degradability (k = 0.031 h^−1^); RUP, rumen undegraded protein; IDP, intestinal digestible protein after 16 h of rumen incubation and 1 h of pepsin and 24 h of pancreatin digestion; IADP, intestinally absorbable digestible protein; TDP, total digestible protein. RUP = 1000 – EDCP (g/kg of CP); IADP = RUP (g/kg of CP) × IDP (g/kg of RUP); TDP = EDCP + IADP.

**Table 5 animals-09-00212-t005:** Pearson’s correlations between chemical composition, ruminal degradability parameters of crude protein, and intestinally absorbable digestible protein of common vetch grains.

Dependent Variable ^1^	CP	EE	NDF	ADF	Ash
*A*	0.581 ^***^	−0.463 ^**^	−0.751^***^	−0.672 ^***^	−0.597 ^***^
*B*	0.605 ^***^	−0.524 ^**^	−0.350 ^*^	−0.461 ^**^	−0.619 ^***^
*C*	–0.611 ^***^	0.549 ^**^	0.533 ^**^	0.620 ^***^	0.710 ^***^
IADP	–0.787 ^***^	0.663 ^***^	0.735 ^***^	0.727 ^***^	0.781 ^***^

^1^*A*, soluble fraction of grain CP; *B*, potentially degradable fraction of grain CP; *C*, rate of degradation of fraction *B*; aNDF, neutral detergent fiber; ADF, acid detergent fiber; CP, crude protein; EE, ether extract; IADP, intestinally absorbable digestible protein. ^*^*p* < 0.05; ^**^
*p* < 0.01; ^***^
*p* < 0.001.

**Table 6 animals-09-00212-t006:** Equations to predict the ruminal degradability parameters *A*, *B*, (g/kg CP), and *C* (h^−1^) of crude protein and intestinally absorbable digestible protein (IADP; g/kg CP) of common vetch grains.

Dependent Variable ^1^	Equation	*R* ^2^	RMSE
*A*	= –1.53 aNDF + 672	0.751	18.8
	= –1.23aNDF – 5.73ash + 777	0.795	17.3
	= –1.38aNDF – 7.18ash + 3.27EE + 794	0.805	16.9
*B*	= –13.9ash + 996	0.619	26.1
	= –8.81ash + 1.33CP + 367	0.676	24.4
	= –8.99ash + 2.01CP + 0.591aNDF – 0.231	0.696	23.8
	= –6.99ash + 2.11CP + 1.22 aNDF – 1.73ADF – 125	0.712	23.3
*C*	= 0.00549ash – 0.0765	0.710	0.00806
	= 0.00409ash + 0.000536ADF – 0.0692	0.748	0.00759
IADP	= –2.51CP + 1076	0.787	17.4
	= –1.55CP + 9.00ash – 465	0.867	14.1
	= –0.828CP + 8.80ash + 0.635aNDF + 70.2	0.891	12.8

^1^ Units: g/kg dry matter for grain chemical composition of CP, EE, aNDF, and ash; *A*, soluble fraction of grain CP; *B*, potentially degradable fraction of grain CP; *C*, rate of degradation of fraction *B* (h^−1^); aNDF, neutral detergent fiber; CP, crude protein; EE, ether extract; *R*^2^, coefficients of determination; RMSE, root mean squared error.

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
