# Peer review of "Comparative Grain Chemical Composition, Ruminal Degradation In Vivo, and Intestinal Digestibility In Vitro of Vicia Sativa L. Varieties Grown on the Tibetan Plateau"

_animals, 2019, doi:10.3390/ani9050212_

Round 1

Reviewer 1 Report

The objectives of the study were to evaluate chemical composition, ruminal degradability kinetics of CP and intestinal protein digestibility of four common vetch varieties over two cropping years. Besides, they assessed the relationship between grain protein, fiber fractions, and ash concentrations, and grain ruminal degradability parameters of CP and intestinally digestible protein of the vetch grain.

General contents:

The topic of this article is interesting and it tries to assess the effect of the CV and year crop on chemical composition and nutritive value of vetch grain for animal nutrition.  The manuscript is well wrote with a good English and good structure.

Keywords, some of them are repeated... it is better if you look for other, to make the search on line wider.. but up to you..

An important doubt:

There is not any interaction between year and variety?? It is very strange

If there are interaction, tables should show them or figures should be drawn.

It is hard to think that no interactions between varieties and year had found.  

Please clearify this point. The rest of manuscript is well presented and discussed.

Author Response

Response to Reviewer 1 Comments

Point 11. Keywords, some of them are repeated... it is better if you look for other, to make the search on line wider.. but up to you.

Response 1: We changed the keywords “common vetch; grain quality; rumen degradability; intestinal digestibility; ruminants” to “common vetch; grain; nutritive value; ruminants”.

Point 2: There is not any interaction between year and variety?? It is very strange.

Response 2: That is correct – there was no significant interaction between year and variety for the dependent variables in this study. We clarified this within the revised manuscript on Lines 212-213, 229-230, 237, 244, 249, 258-259, 265 and Table 3 and 4. This can be explained by the finding that in both years, differences in grain nutrient composition among varieties were consistent among years. Similar results have been reported for nutrient content of whole cottonseed (Bertrand et al., 2005; utrient Content of Whole Cottonseed) and chemical composition and nutritive value of pea seed (Kotlarz et al., 2011; Chemical composition and nutritive value of protein of the pea Seeds - effect of harvesting year and variety).

Additional minor revisions were made to the manuscript using track changes to improve English and readability, based on suggestions from our co-author whose native language is English.

Reviewer 2 Report

Title: I suggest to change the title underline the in vivo and in vitro used methods “Comparative grains chemical composition, ruminal degradation in vivo and intestinal digestibility in vitro of Vicia Sativa L. varieties grown on the Tibetan plateau”

Line 16: change in “few informations are”

Line 19: add “in vivo and in vitro”

Line 77: please, specify the ban of the use of soybean meal in organic livestock.” Recently some obstacles are limiting the use of soya bean: the ban in organic livestock (EC Council Regulation 834/2007) due to the chemical treatment and its costs and availability strongly related with the price development of agricultural commodities on the world market” citing this article “Musco N., et al. Journal of Animal Physiology and Animal Nutrition, 101(6): 1227-1241”.

Line 77: ruminant? Please specify the specie.

Line 83: change instead of “animal species” “the species to which it is administered”.

Line 95: add a reference.

Line 126: please describe also the “local variety” in terms of cultivar, origin and its spread in the area.

Chemical analysis: please report a more recent reference of AOAC, and the ID methods for each performed analysis.

Line 158: please specify more details of the animals used in the in vivo trial. Mean age, sex, body weight, basal diet.

Line 170: the trial authorization number MUST BE specified.

Table 3: remove letter “a” from CP.

Line 215: please add phosphorus (P).

Line 272-275: this period must be reported in “statistical analysis section”

Line 296: help TO explain.

Line 319: showed that.

Check the references: not all of there are reported according to the guidelines of the Journal.

Author Response

Response to Reviewer 2 Comments

Point 1:Title:  I suggest to change the title underline the in vivo and in vitro used methods “Comparative grains chemical composition, ruminal degradation in vivo and intestinal digestibility in vitro of Vicia Sativa L. varieties grown on the Tibetan plateau”

Response 1: Based on the suggestions from Reviewer 2, with minor editorial suggestions from our co-author whose native language is English, we changed the title from “Comparative grain chemical composition, ruminal degradation and intestinal digestibility of four common vetch varieties grown on the Tibetan plateau” to “Comparative grain chemical composition, ruminal degradation in vivo, and intestinal digestibility in vitro of Vicia Sativa L. varieties grown on the Tibetan plateau”.

Point 2:Line 16: change in “few informations are”

Response 2: We have changed “little information is” to “few informations are”.  Please see in Line 16.

Point 3Line 19: add “in vivo and in vitro”

Response 3: We added “in vivo” and “in vitro” within lines 19-20 of the revised manuscript.

Point 4Line 77: please, specify the ban of the use of soybean meal in organic livestock.” Recently some obstacles are limiting the use of soya bean: the ban in organic livestock (EC Council Regulation 834/2007) due to the chemical treatment and its costs and availability strongly related with the price development of agricultural commodities on the world market” citing this article “Musco N., et al. Journal of Animal Physiology and Animal Nutrition, 101(6): 1227-1241”.

Response 4: We added “due to the banning of soybean (Glycine max (L.) Merr.) meal for feeding organic livestock” and the reference (Musco et al., 2017) within lines 79-80 of the revised manuscript.

Point 5: Line 77: ruminant? Please specify the specie.

Response 5: We added “such as cattle and sheep” within line 78 of the revised manuscript.

Point 6: Line 83: change instead of “animal species” “the species to which it is administered”.

Response 6: We changed “from single specie” to “single variety” within line 84 of the revised manuscript.

Point 7: Line 95: add a reference.

Response 7: We added the references “[21, 22]” within line 98 of the revised manuscript.

Point 8: Line 126: please describe also the “local variety” in terms of cultivar, origin and its spread in the area.

Response 8: We added the sentence “The local variety originated in Gansu province of China and is now widely cultivated in Gansu, Qinghai, and provinces located in the middle and lower reaches of the Yangtze River in China.” within lines 130-132 of the revised manuscript.

Point 9: Chemical analysis: please report a more recent reference of AOAC, and the ID methods for each performed analysis.

Response 9: We added this information within lines 150-159 of the revised manuscript.

Point 10: Line 158: please specify more details of the animals used in the in vivo trial. Mean age, sex, body weight, basal diet.

Response 10: We added this information according to referee’s comments. Please see in Line 162-163; Line 175.

Point 11: Line 170: the trial authorization number MUST BE specified.

Response 11: We have made a revision according to referee’s comments. Please see Line 178-180.

Point 12: Table 3: remove letter “a” from CP.

Response 12: We removed “a” behind “CP” within the first column of Table 3 of the revised manuscript. For enhanced readability of the revised manuscript, this change was made with the track changes feature of Microsoft Word turned off.

Point 13: Line 215: please add phosphorus (P).

Response 13: We have made a revision according to referee’s comments.

Point 14: Line 272-275: this period must be reported in “statistical analysis section”.

Response 14: We added this information within lines 196-199 of the revised manuscript (the statistical analyses section). We also added additional detail regarding calculation of RUP, IADP, and TDP within lines 204-206 of the revised manuscript (the statistical analyses section) for enhanced clarity.

Point 15: Line 296: help TO explain.

Response 15: As suggested by the reviewer, we added “to” within line 313 of the revised manuscript.

Point 16: Line 319: showed that.

Response 16: As suggested by the reviewer, we added “that” within line 337 of the revised manuscript.

Point 17: Check the references: not all of there are reported according to the guidelines of the Journal.

Response 17: We have made a revision according to referee’s comments. Please see Line 462-465; Line 472-473; Line 480-481; Line 486-487.

Additional minor revisions were made to the manuscript using track changes to improve English and readability, based on suggestions from our co-author whose native language is English.

Special thanks to you for your good comments. We hope that the correction will meet with approval.

Round 2

Reviewer 2 Report

Dear authors,

the Paper 484225 entitled "Comparative grain chemical composition, ruminal degradation in vivo, and intestinal digestibility in vitro of Vicia Sativa L. varieties grown on the Tibetan plateau" is now suitable for publication. All the requests of the reviewer process were met by the authors, also the English language was improved. In my opinion can be accepted in the present form.